# Differences between Motile and Nonmotile Cells of *Haematococcus pluvialis* in the Production of Astaxanthin at Different Light Intensities

**DOI:** 10.3390/md17010039

**Published:** 2019-01-09

**Authors:** Feng Li, Minggang Cai, Mingwei Lin, Xianghu Huang, Jun Wang, Hongwei Ke, Xuehong Zheng, Ding Chen, Chunhui Wang, Shaoting Wu, Yu An

**Affiliations:** 1The Fujian Provincial Key Laboratory for Coastal Ecology and Environmental Studies, Xiamen University, Xiamen 361101, China; lifeng@stu.xmu.edu.cn; 2Coastal and Ocean Management Institute, Xiamen University, Xiamen 361101, China; 3Key Laboratory of Marine Chemistry and Applied Technology, Xiamen 361101, China; xhzheng@xmu.edu.cn (X.Z.); chending@xmu.edu.cn (D.C.); springfl@xmu.edu.cn (C.W.); 4College of Ocean and Earth Science, Xiamen University, Xiamen 361101, China; 22320142200997@stu.xmu.edu.cn (M.L.); Hongwei_KE@xmu.edu.cn (H.K.); 22320142201088@stu.xmu.edu.cn (S.W.); 22320142201072@stu.xmu.edu.cn (Y.A.); 5Xiamen Ocean Vocational College, Xiamen 361101, China; wangjun@mail.maritech.com.cn; 6College of fisheries, Guangdong Ocean University, Zhanjiang 524088, China; hxh166@126.com

**Keywords:** *Haematococcus pluvialis*, astaxanthin, light intensity, photooxidative stress

## Abstract

*Haematococcus pluvialis*, as the best natural resource of astaxanthin, is widely used in nutraceuticals, aquaculture, and cosmetic industries. The purpose of this work was to compare the differences in astaxanthin accumulation between motile and nonmotile cells of *H. pluvialis* and to determine the relationship between the two cells and astaxanthin production. The experiment design was achieved by two different types of *H. pluvialis* cell and three different light intensities for an eight day induction period. The astaxanthin concentrations in nonmotile cell cultures were significantly increased compared to motile cell cultures. The increase of astaxanthin was closely associated with the enlargement of cell size, and the nonmotile cells were more conducive to the formation of large astaxanthin-rich cysts than motile cells. The cyst enlargement and astaxanthin accumulation of *H. pluvialis* were both affected by light intensity, and a general trend was that the higher the light intensity, the larger the cysts formed, and the larger the quantity of astaxanthin accumulated. In addition, the relatively low cell mortality rate in the nonmotile cell cultures indicated that the nonmotile cells have a stronger tolerance to photooxidative stress. We suggest that applying nonmotile cells as the major cell type of *H. pluvialis* to the induction period may help to enhance the content of astaxanthin and the stability of astaxanthin production.

## 1. Introduction

Astaxanthin is a high-value red ketocarotenoid of a predominantly marine origin with a powerful antioxidant capacity and anti-inflammatory property [1,2,3]. As one of the promising agents for the prevention of oxidative stress-related diseases, astaxanthin has received extensive attention in the basic and clinical research area in recent years [4,5,6]. Natural astaxanthin is widely found in various microorganisms and marine animals [7,8]. Among them, the microalgae *Haematococccus pluvialis* (Chlorophyceae, Volvocales) is considered to be the best natural resource for the commercial production of natural astaxanthin because it can synthesize and accumulate natural astaxanthin to 3–5% of its own dry weight [9,10,11].

*H. pluvialis* is considerable better suited for survival under different bodies of water than most algae because of its rapid ability to encystment, especially inhabits coastal rocks near the seaside with conditions of expeditious fluctuations in light, temperature, and salt concentration [12,13,14]. *H. pluvialis* has evolved a complex cell transformation strategy, in which four types of cells are distinguished: microzooids, macrozooids, nonmotile palmella cells, and haematocysts (aplanospores), to cope with and survive under these fluctuation conditions [15,16]. The microzooids and macrozooids are also described as motile cells which can be driven by two isometric flagella [17,18]. The motile cell grows primarily in favorable environmental conditions, such as plenty of nutrients, and an adequate temperature and light intensity [10,19]. When growing environment or culture conditions become less favorable, the motile cell may lose its flagella and develop into a spherical nonmotile form, which is also called the ‘palmella’ stage [16]. Both motile and nonmotile cells are defined as vegetative cells [17]. When environmental conditions further become unfavorable for growth, *H. pluvialis* cells accumulate astaxanthin during the transformation from vegetable cells to cyst cells, which is considered an adaptation of this organism to adverse environments [20,21].

The widely adopted strategy for the production of *H. pluvialis* astaxanthin is a “two-stage” batch culture mode, which first produces green vegetative cells under favorable conditions to obtain enough biomass accumulation (“green stage”) and then exposes the cells to stress environmental conditions to induce astaxanthin production (“red stage”) [22,23,24]. Although *H. pluvialis* has achieved commercial application, several investigations have reported that a large amount of cell death among vegetable cells occurred during the first few days after transferring *H. pluvialis* from the green stage to the red stage, resulting in the overall astaxanthin productivity being low [25,26,27]. The cell mortality rate of motile and nonmotile vegetable cells may be different due to the differences in the tolerance of them to stress [17]. Therefore, it is speculated that there may be differences in the production of astaxanthin between the motile and nonmotile cells of *H. pluvialis*. However, little information is available on the differences in the astaxanthin accumulation between motile and nonmotile cells of *H. pluvialis* under stress conditions. Furthermore, the relationship between the cell types of *H. pluvialis* and astaxanthin content has not been determined.

In the present study, we investigated astaxanthin production using two different types of cells (motile and nonmotile cells) of *H. pluvialis* at different light intensities. The purpose of this work was to compare the differences in astaxanthin content and cell mortality rate between the two cells and to determine the relationship between the two cells and astaxanthin accumulation.

## 2. Results

### 2.1. The Accumulation of Biomass and Astaxanthin

To determine the differences in the production of biomass and astaxanthin between motile and nonmotile cells at different light intensity conditions, we selected 30, 80, and 150 μmol·m^−2^·s^−1^, representing low, medium, and high light intensities, respectively, to conduct experiments. As shown in Figure 1a–c, the biomass concentration in nonmotile cell cultures showed a linear increase and the trend was that the higher the light intensity, the greater the increase in biomass concentration. In contrast, a decrease in biomass concentrations after day 5 was observed in motile cell cultures at the light intensity of 80 and 150 μmol·m^−2^·s^−1^. The concentrations of biomass increased by 28% and 44% in motile and nonmotile cell cultures, respectively, when increasing the light intensity from 30 μmol·m^−2^·s^−1^ to 150 μmol·m^−2^·s^−1^. The concentration of biomass was higher in nonmotile cell cultures in comparison to motile cell cultures.

As shown in Figure 1d–f, a linear increase in astaxanthin concentration was observed in all the *H. pluvialis* cultures and a general trend was that the higher the light intensity, the more rapid the rise in astaxanthin occurred. The maximum concentrations of astaxanthin reached 23.93 ± 0.98 mg·L^−1^ and 82.82 ± 3.29 mg·L^−1^ in motile and nonmotile cell cultures, respectively. Significant differences in astaxanthin concentrations were observed between motile and nonmotile cell cultures at the same light intensity. The astaxanthin concentrations in nonmotile cell cultures increased by 80%, 185% and 246% compared to motile cell cultures at 30, 80 and 150 μmol·m^−2^·s^−1^, respectively.

As shown in Figure 1g–i, the initial astaxanthin content in dry weight was higher in motile cell cultures in comparison to nonmotile cell cultures. When the light intensity was 80 μmol·m^−2^·s^−1^ or higher, a decline of astaxanthin content occurred within the first two days in motile cell cultures, followed by a gradual increase in astaxanthin content. A rapid increase in astaxanthin content was observed within the first four or six days in nonmotile cell cultures, and the trend was that the higher the light intensity, the greater the increase in astaxanthin content. Furthermore, it was observed that the differences in astaxanthin content between motile and nonmotile cell cultures increased with increasing light intensity. When the light intensity increased to 150 μmol·m^−2^·s^−1^, the maximum astaxanthin content reached 3.9% Dry weight (DW) in nonmotile cell cultures, which was 1.64 times that of motile cells cultures.

### 2.2. The Productivity of Biomass and Astaxanthin

As shown in Figure 2a–c, the maximum biomass productivity of both cell types increased with increasing light intensity. At the light intensity of 30 μmol·m^−2^·s^−1^, the maximum biomass productivity in both types of cells was obtained on day 2 of the induction period, followed by a gradual decline in biomass productivity (Figure 2a). When the light intensity increased to 80 μmol·m^−2^·s^−1^ or higher, the productivity of biomass in both types of cell cultures decreased with the increase of time during the induction period and the rate of decrease in biomass productivity increased as the light intensity increased. Under the same light intensity, the rate of decline in biomass productivity was faster in motile cells in comparison to nonmotile cells. After eight days of induction at the light intensity of 80 μmol·m^−2^·s^−1^ and 150 μmol·m^−2^·s^−1^, the biomass productivity decreased by 69% and 74%, respectively, in motile cell cultures, corresponding to 41% and 45% in nonmotile cell cultures.

The astaxanthin productivities of motile and nonmotile cell cultures at different light intensities are shown in Figure 2d–f. A significant effect of light intensity on the astaxanthin productivity of nonmotile cells was observed compared to motile cells. When the light intensity was 30 μmol·m^−2^·s^−1^ and 80 μmol·m^−2^·s^−1^, the astaxanthin productivity of nonmotile cell cultures increased in a linear fashion (Figure 2d,e). When the light intensity was 150 μmol·m^−2^·s^−1^, the astaxanthin productivity of nonmotile cell cultures increased to the maximum level on day 4 of induction, and then the value declined. The maximum astaxanthin productivity values of 3.6, 5.3 and 11.8 mg·L^−1^·day^−1^ were obtained in nonmotile cell cultures, respectively, which was 1.9-, 2- and 4.2-fold that in motile cell cultures at the same light intensity.

The protective roles of astaxanthin biosynthesis can reduce the effect of oxidative stress on the cells to a certain extent [20,28,29,30]. Therefore, we used the astaxanthin content increase rate to evaluate the extent to which the cells defend themselves under high light intensity stress conditions. As shown in Figure 2g–i, little difference in the astaxanthin content increase rate was observed between motile and nonmotile cell cultures at the light intensity of 30 μmol·m^−2^·s^−1^. When the light intensity was 80 and 150 μmol·m^−2^·s^−1^, the maximum astaxanthin content increase rates of 0.6% DW·day^−1^ and 0.8% DW·day^−1^, respectively, were obtained in nonmotile cell cultures on day 2 of induction, followed by a gradual decrease in the astaxanthin content increase rate. In contrast, the astaxanthin content increase rate in biomass showed a negative increase in motile cells culture during the first two days of induction, and then increased gradually. The large differences in the astaxanthin content increase rate between the two cells in the early stage of induction indicated that once exposed to photooxidative stress, the nonmotile cells can rapidly synthesize and accumulate astaxanthin in a short time.

### 2.3. The Cell Morphology and Viability 

Microscopic examination showed that there were changes in the cell morphology during the induction period, where green cells turned into red cysts, accompanied by the accumulation of astaxanthin (Figure 3a–c). The motile cells were green at first, and ellipsoidal or pear-shaped with two flagella. As the induction time increased, the motile cells lost their flagella and developed into red cysts with a thick cell wall. At the early stage of induction, red pigmentation due to astaxanthin accumulation appeared towards the center of the green nonmotile cells. The red pigmentation gradually occupied the entire cell volume, resulting in the formation of red large cysts. A significant effect of light intensity on the process of red cysts transformation of nonmotile cells showed that the higher the light intensity, the earlier the red pigmentation occurred and the larger the quantity of astaxanthin accumulated in the cells. Furthermore, it was observed during the induction period that the nonmotile cells accumulated significantly more astaxanthin than the motile cells under a given light intensity.

Cell death, which showed an upward trend with an increasing light intensity, in all cultures, was observed during the induction period (Figure 3a–c). As shown in Table 1, the cell mortality rate in motile cell cultures increased to ca. 22% from 8% when increasing the light intensity from 30 to 80 μmol·m^−2^·s^−1^. When the light intensity increased to 150 μmol·m^−2^·s^−1^, the value of the cell mortality rate reached ca. 28% in motile cell cultures. Compared to motile cells, the value of the mortality rate in nonmotile cell cultures was much lower under the same light intensity condition. The cell mortality rate of nonmotile cell cultures was ca. 4%, 7% and 8%, respectively, at the light intensity of 30, 80 and 150 μmol·m^−2^·s^−1^.

### 2.4. The Diameter of Cysts

Changes in the size of the cysts formed by motile and nonmotile cells were observed. The size of cysts from motile cell cultures reached an average diameter of 19.76 ± 2.66, 21.38 ± 3.44, and 21.39 ± 3.85 μm, respectively, at the light intensity 30, 80 and 150 μmol·m^−2^·s^−1^ (Table 2). A significantly increase of the diameter of cysts was observed in nonmotile cells compared with motile cells. The average value of cysts’ diameter of 28.71 ± 5.22, 33.07 ± 4.25 and 33.97 ± 5.35 μm was obtained in nonmotile cell cultures, respectively, at the three different light intensity conditions.

When the influences of different light intensity on the size of *H. pluvialis* cysts formed by two cells were considered, the distribution of the cysts in different diameter ranges was evaluated in cultures. As shown in Figure 4a–c, the diameter of cysts was mainly distributed in the range of 10–30 μm in the motile cell cultures, while the diameter of cysts formed by nonmotile cells was mainly distributed in the range of 20–50 μm. Under the light intensity of 30 μmol·m^−2^·s^−1^, ca. 40% of cysts ranged in the diameter of 20–30 μm in motile cell cultures. When increasing the light intensity to 80 μmol·m^−2^·s^−1^ and 150 μmol·m^−2^·s^−1^, the percentage of cysts which had a diameter greater than 20 μm in motile cell cultures increased to ca. 62% and 64%, respectively. In nonmotile cell cultures, after eight days of being illuminated at 30, 80 and 150 μmol·m^−2^·s^−1^, the proportion of cysts in the diameter range of greater than 30 μm reached 45%, 76% and 79%, respectively.

To determine the relationship between the cell size and astaxanthin content of red cysts, we calculated the average biomass and astaxanthin content of individual cyst cells in cultures at different light intensities. The results in Table 2 indicated that the increase in biomass and astaxanthin content in cyst cells was closely associated with the enlargement of cell size, and both were affected by light intensity. The trend was that the higher the light intensity, the larger the quantity of astaxanthin accumulated, and the larger the cysts size formed. In addition, under the same light intensity, the size of cysts formed by nonmotile cells was bigger than that formed by motile cells.

## 3. Discussion

Most of the previous studies focused on the production of astaxanthin using *H. pluvialis* vegetable cells [24,31,32,33,34], which are a mixture usually composed of green motile vegetative cells and green nonmotile vegetative cells. The green motile vegetative cells represent a group of young swimming cells driven by two flagella, including zoospores which come from the asexual reproduction of *H. pluvialis*. When motile cells aged, they lost their flagella to form spherical nonmotile cells (palmelloid) [34,35,36], which can efficiently utilize the chemical energy accumulated during the process of the transformation of motile cells to nonmotile cells to help themselves rapidly synthesize and accumulate astaxanthin under stress conditions [9,17]. The ratio of motile cells to nonmotile cells in green vegetative cells at different culture times is different, which may affect the accumulation of astaxanthin in the red stage, thereby resulting in the instability of astaxanthin production. Sarada et al. [37] observed that older *H. pluvialis* cells (12–16 days old) accumulated more astaxanthin than younger cells (4–8 days old). Han et al. [17] suggested that nonmotile cells have an advantage over flagellated motile cells for the production of astaxanthin. In this study, we confirmed that the biomass and astaxanthin concentrations in nonmotile cell cultures were 0.88 to 1.15 times and 0.80 to 2.46 times higher than those in motile cell cultures at different light intensities conditions, respectively. Comparatively, the nonmotile cells of *H. pluvialis* were more suitable as the main cell type for the induction of *H. pluvialis* to product astaxanthin in the red stage due to their higher biomass and astaxanthin productivity. 

Light intensity is considered as one of the most important environmental factors affecting the accumulation of astaxanthin in *H. pluvialis*. The high light intensity can cause large quantities of astaxanthin biosynthesis and accumulation in *H. pluvialis* [27,31,38,39,40]. Kobayashi et al. [38] suggested that the astaxanthin content proportionally correlated to light quantity (light intensity × net illumination time). A significant effect of high light intensity on the level of astaxanthin accumulated in *H. pluvialis* cells was observed by Harker et al. [31]. Recently, Li et al. [27] reported that the higher the photon flux density, the larger the quantity of astaxanthin accumulated in the cells. However, excess reactive oxygen species (ROS) which can potentially react with major macromolecules (e.g., DNA, lipids and protein), were produced in cells when exposed to a high light condition, resulting in cellular damage [41,42,43]. The cell mortality rate may range from 13% to 82% of total cells in different strains of *H. pluvialis* at different light intensities. Hu et al. [26] reported that the highest cell mortality rate of *H. pluvialis* NIES-144 strains reached ca. 40% at the light intensity 250 μmol·m^−2^·s^−1^, while Hata et al. [25] and Li et al. [27] used the same strains to obtain a ca. 70% and 82% cell mortality rate under the light intensity of 950 μmol·m^−2^·s^−1^ and 600 μmol·m^−2^·s^−1^, respectively. However, these investigations carried out used green vegetative cells. In this study, the mortality rate of 8% was obtained in the nonmotile cell cultures at 150 μmol·m^−2^·s^−1^, which was the lowest reported in the induction of *H. pluvialis* by high light intensity (Table 1). The reasons for the relatively low mortality rate of nonmotile cells may be that nonmotile cells not only reduce the cytochrome f level via down-regulating the linear electron transport chain, but also enhance the plastid terminal oxidase (PTOX) pathway to consume excess electrons produced by PSII [17,44]. In addition, we confirmed that once exposed to photooxidative stress, the nonmotile cells can rapidly synthesize and accumulate astaxanthin in a short time, and the increased astaxanthin may act as ‘sunscreen’ to reduce chloroplast-based ROS formation under high light stress [20,21], thereby reducing cellular damage.

The changes in the color, shape, and size of *H. pluvialis* cells are a reflection of the adaptation of this organism to adverse environments. In this study, we confirmed that the increase of astaxanthin content of *H pluvialis* cyst cells is closely associated with the enlargement of cyst size, and both are affected by light intensity (Table 2). When exposed to photooxidative stress, *H. pluvialis* cells began to synthesize and accumulate astaxanthin to defend themselves from photooxidative damage [20,30,45]. During the process of astaxanthin accumulation, an increase in lipid content and carbohydrate was observed in previous investigations [9,34]. The increased carbohydrate may be used to support the synthesis of fatty acids, which may serve as a matrix for solubilizing the astaxanthin ester [10,46]. Zhekisheva et al. [47] suggested that the high content of astaxanthin requires a large amount of oleic acid accumulation to maintain it. Thus, the increased compounds during astaxanthin accumulation required more space for storage. This may account for the enlargement of cysts. Due to motile cells being more susceptible to high light than nonmotile cells [17], cell damage occurred once exposed to high light, resulting in a relatively low astaxanthin content in motile cells under photooxidative stress, and this may further influence the enlargement of cysts. In contrast, under photooxidative stress, nonmotile cells more effectively converted fixed photochemical energy into storage neutral lipids than motile cells and accompanied by the accumulation of astaxanthin [17], as a result, the size and mass of red cyst increased. In addition, the distribution of the red cysts’ diameter in different ranges indicated that the large red cysts of *H. pluvialis* might mainly be formed by nonmotile cells, which is important for understanding the survival strategy of *H. pluvialis* in adverse environments.

In the present study, we have determined the importance of *H. pluvialis* nonmotile cells for astaxanthin production. We show that the astaxanthin and biomass concentrations and cyst cells’ size are significantly increased in the nonmotile cell cultures compared with motile cells. The analysis results of the relationship between the cell size and astaxanthin content of red cysts indicate that the increase of astaxanthin is closely associated with the enlargement of cell size, and the nonmotile cells are more conducive to the formation of large astaxanthin-rich cysts than motile cells. The effect of the different light intensities on astaxanthin accumulation and cysts’ enlargement confirms that the higher the light intensity, the faster the red pigmentation appeared, the larger the quantity of astaxanthin accumulated in cells, and the larger the cysts’ size formed. In addition, cell death occurred with the increase in light intensity, and the relatively low mortality of the nonmotile cell cultures indicates that the nonmotile cell is able to cope with and survive under photooxidative stress. These results provide further evidence that applying nonmotile cells as the major cell type of *H. pluvialis* to the induction period might help a *H. pluvialis* culturist to improve the existing strategy which produces astaxanthin using vegetable cells of *H. pluvialis*, in order to enhance the content of astaxanthin and the stability of astaxanthin production. These results might also help us to more effectively understand the survival strategies of the different cell types of *H. pluvialis* under photooxidative stress and other adverse environmental conditions.

## 4. Materials and Methods 

### 4.1. H. pluvialis Strain and Growth Conditions

*H. pluvialis* CCMA-451 (Genbank accession number MG847145) was from the Yellow sea coastal rocks (Qingdao, China) and was preserved in the Center for Collections of Marine Algae at Xiamen University, Xiamen, China. Stock cultures of *H. pluvialis* were maintained at 15 μmol·m^−2^·s^−1^ in Bold Basal Medium. Motile cells were cultivated autotrophically in liquid Bold Basal Medium with 0.75 g·L^−1^ NaNO_3_ at 25 ± 1 °C under a continuous low light intensity of 30 μmol·m^−2^·s^−1^ in 3 L flasks for eight days. A part of the five-day-old motile cell culture was diluted with phosphate-depleted medium containing 1.25 g·L^−1^ NaCl at a ratio of 1:4, followed by cultivating under 30 μmol·m^−2^·s^−1^ for three days to prepare nonmotile cells. To increase the quantity of nonmotile cells, cells that settled at the bottom of the glass columns were collected and washed with fresh aseptically medium several times to remove remaining motile cells.

### 4.2. Stress Conditions

The experiments were performed in a 1 L glass column (inner diameter 5 cm). The cells from motile and nonmotile cultures were collected and concentrated by centrifugation (2000 rpm, 2 min) and removed the supernatant. The collected cells were inoculated into 600 ml sterilized N and P depletion medium in a 1 L glass column to achieve the initial cell concentration of 0.5 × 10^6^ cell·mL^−1^. The experimental design was achieved by two different types of cell and three different light intensities at 25 ± 1 °C for an eight days induction period. The cell types were motile cells and nonmotile cells. The two cells were tested at three light intensities: 30, 80, and 150 μmol·m^−2^·s^−1^, representing a low, medium, and high light, respectively. All cultures were mixed by continuous bubbling with 1.5% (*v*/*v*) CO_2_. Illumination was provided from the side by 100% red (620–630 nm), 100% green (520–530 nm), and 100% blue (465–475 nm) LED plant grow lights (Xiamen Top-Succeed Photobiology Technology Co., Ltd, Xiamen, China). The experiments were simultaneously repeated three times under the same stress conditions.

### 4.3. Analytical Methods

Dry weight (DW) was determined daily by filtering a 10 mL culture sample of the algal suspension through pre-weighed (m_1_) Whatman GF/C filter paper (47 mm, 1.2 μm). The filter paper was dried at 90 °C for 500 min to a constant weight and weighed with microbalance (m_2_). The DW was calculated with Equation (1).
(1)$$\mathrm{DW}\;\left(g\cdot L^{-1}\right)=\left[m_{2}\;\left(g\right)-m_{1}\;\left(g\right)\right]\times 10^{3}/10\;\left(\mathrm{mL}\right)$$

Biomass productivity was calculated with Equation (2).
(2)$$\mathrm{Biomass}\;\mathrm{productivity}\;\left(g\cdot L^{-1}\cdot {\mathrm{day}}^{-1}\right)=\left[{\mathrm{DW}}_{t}\;\left(g\cdot L^{-1}\right)-{\mathrm{DW}}_{0}\;\left(g\cdot L^{-1}\right)\right]/t\;\left(\mathrm{day}\right)$$
where DW_t_ and DW_0_ were the dry weight of day t and day 0, respectively.

The astaxanthin concentration was determined photometrically [48]. A 10 mL culture sample was collected by centrifugation at 7000 rpm for 5 min, and the pellet was first treated with 5 mL solution of 5% (*w*/*v*) KOH in 30% (*v*/*v*) methanol in a 75 °C water bath for 10 min to remove the chlorophyll. The remaining pellet was then extracted with DMSO after adding 25 μL acetic acid at 75 °C for 10 min. This last step was repeated several times until the sample was colorless and to recover the astaxanthin. The absorbance of the combined extracts was measured at 492 nm ($E_{1cm}^{1\%}$ = 2220) [9], and the astaxanthin concentration was calculated with Equation (3).
(3)$$\mathrm{AX}\;\left(\mathrm{mg}\cdot L^{-1}\right)=A_{492}\times 1000/\left(E_{1cm}^{1\%}\times 100\right)\times V_{a}\;\left(\mathrm{mL}\right)/10\;\left(\mathrm{mL}\right)\times f$$
where AX was the astaxanthin concentration, A_492_ was the absorbance of extracts at 492 nm, V_a_ was the volume of extracts, and *f* was the dilution ratio of measuring the absorbance.

Astaxanthin productivity was calculated with Equation (4).
(4)$$\mathrm{Astaxanthin}\;\mathrm{productivity}\;\left(\mathrm{mg}\cdot L^{-1}\cdot {\mathrm{day}}^{-1}\right)=\left[{\mathrm{AX}}_{t}\;\left(\mathrm{mg}\cdot L^{-1}\right)-{\mathrm{AX}}_{0}\;\left(\mathrm{mg}\cdot L^{-1}\right)\right]/t\;\left(\mathrm{day}\right)$$
where AX_t_ and AX_0_ were the astaxanthin concentration of day t and day 0, respectively.

Astaxanthin content was calculated with Equation (5).
(5)$$C\;\left(\%\right)={\mathrm{AX}}_{t}\left(\mathrm{mg}\cdot L^{-1}\right)\times 10^{-3}/{\mathrm{DW}}_{t}\;\left(g\cdot L^{-1}\right)\times 100\%$$
where C, AX_t_, and DW_t_ were the astaxanthin content, astaxanthin concentration, and dry weight, respectively, on day t.

The astaxanthin content increase rate was calculated with Equation (6).
(6)$$\mathrm{Astaxanthin}\;\mathrm{content}\;\mathrm{increase}\;\mathrm{rate}\;\left(\%\cdot {\mathrm{day}}^{-1}\right)=\left[C_{t}\left(\%\right)-C_{0}\left(\%\right)\right]/t\;\left(\mathrm{day}\right)$$
where C_t_ and C_0_ were the astaxanthin content on day t and day 0, respectively. 

For the observation of the morphological changes, the cells were examined daily using a Leica DM750 light microscope and photos were taken with a Leica ICC50 W camera. The diameter of red cysts on day 8 was determined by using Leica application software with an internal reticle scale.

## Figures and Tables

**Figure 1 marinedrugs-17-00039-f001:**
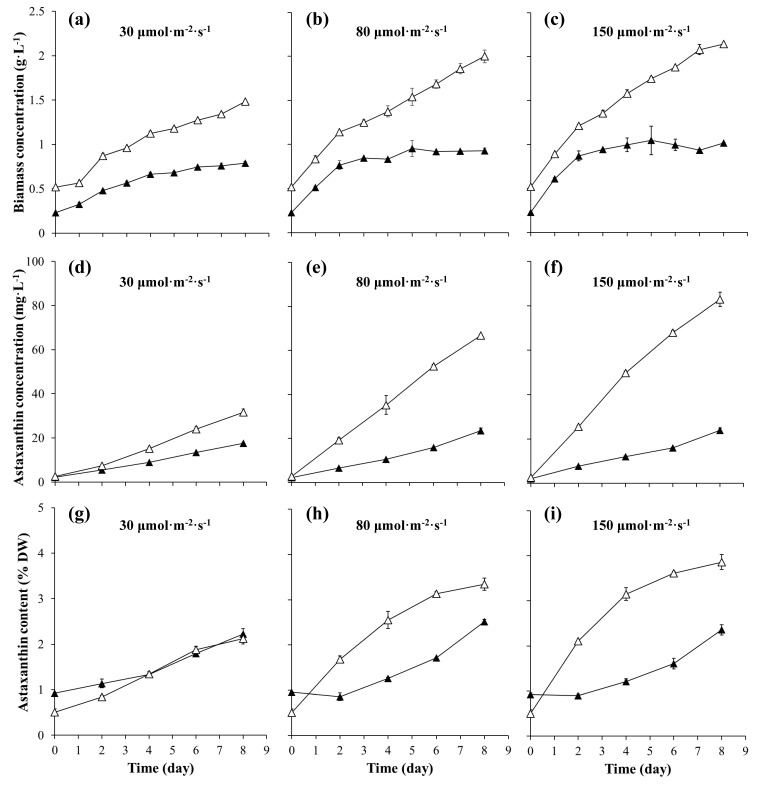
The biomass concentration (**a**–**c**), astaxanthin concentration (**d**–**f**), and astaxanthin content (**g**–**i**) of *H. pluvialis* cultures at different light intensities in the motile cell cultures (black triangle) and the nonmotile cell cultures (white triangle).

**Figure 2 marinedrugs-17-00039-f002:**
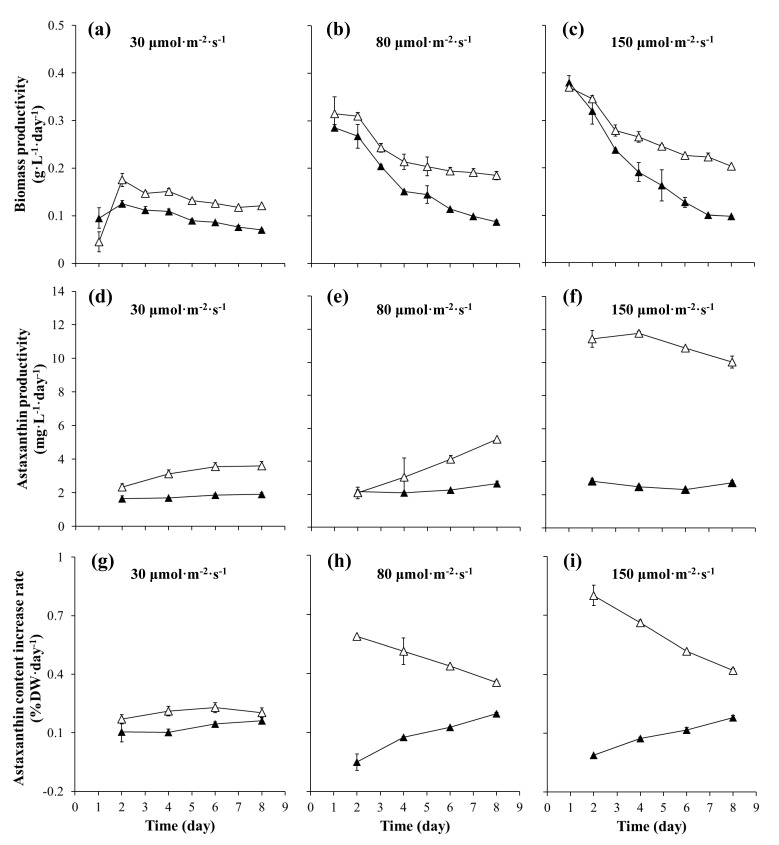
The biomass productivity (**a**–**c**), astaxanthin productivity (**d**–**f**), and astaxanthin content increase rate (**g**–**i**) of *H. pluvialis* cultures at different light intensities in the motile cell cultures (black triangle) and the nonmotile cell cultures (white triangle).

**Figure 3 marinedrugs-17-00039-f003:**
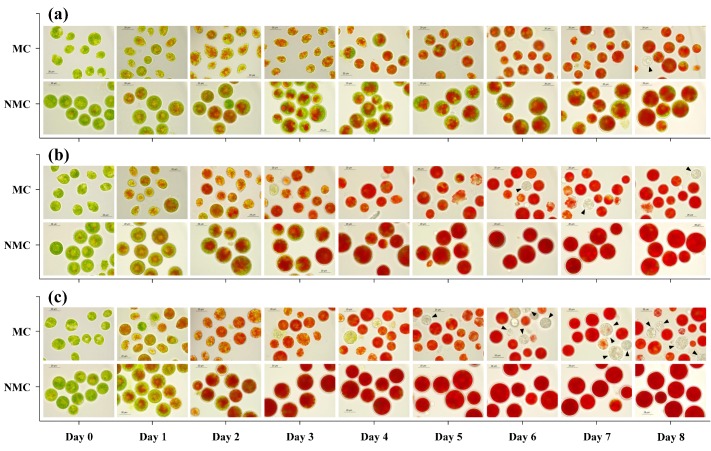
The daily morphological changes of *H. pluvialis* cells at different light intensities during the induction period in the motile cell cultures (MC) and the nonmotile cell cultures (NMC): (**a**) 30 μmol·m^−2^·s^−1^; (**b**) 80 μmol·m^−2^·s^−1^; (**c**) 150 μmol·m^−2^·s^−1^. The dead cells are indicated by arrows.

**Figure 4 marinedrugs-17-00039-f004:**
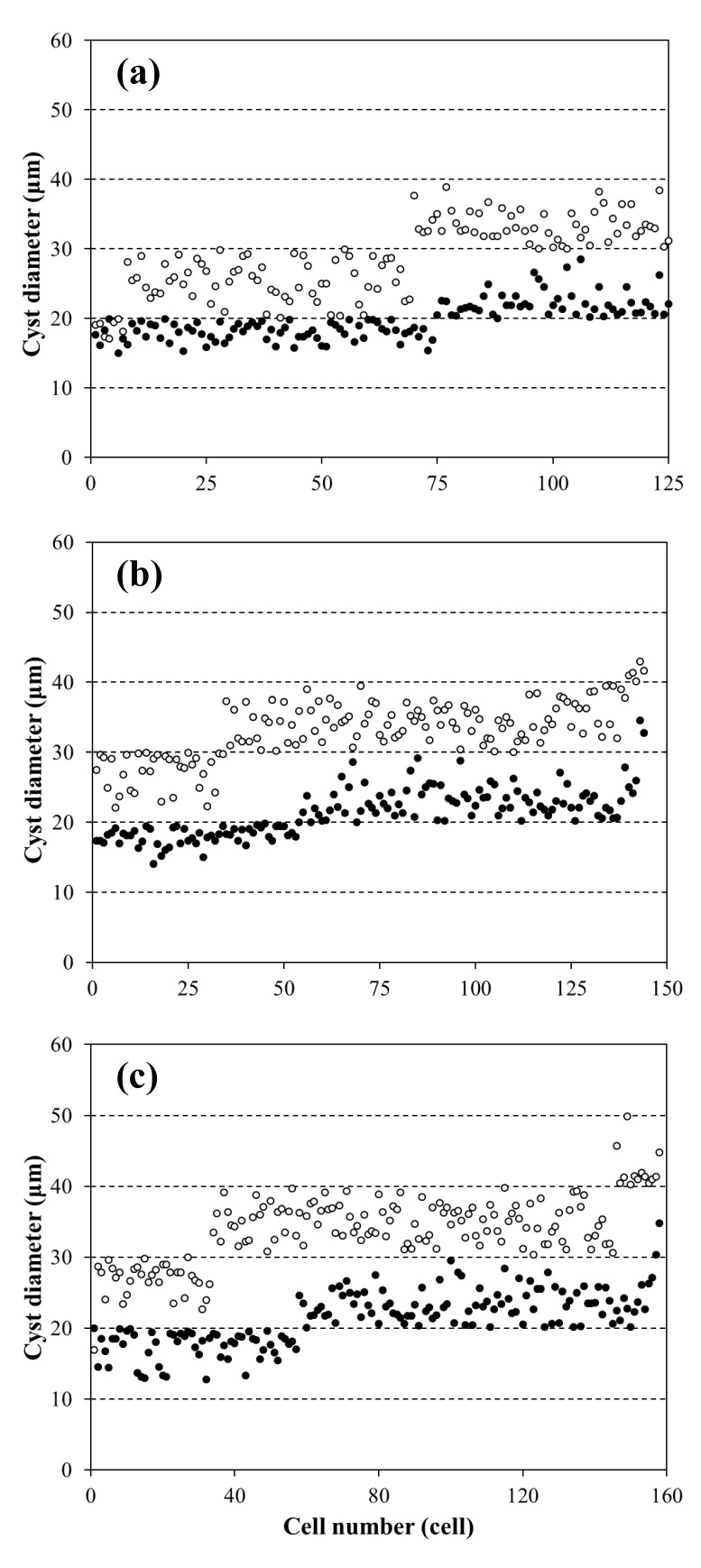
The distribution of *H. pluvialis* cysts in different diameter ranges at different light intensities in the motile cell cultures (black circle) and the nonmotile cell cultures (white circle): (**a**) 30 μmol·m^−2^·s^−1^; (**b**) 80 μmol·m^−2^·s^−1^; (**c**) 150 μmol·m^−2^·s^−1^.

**Table 1 marinedrugs-17-00039-t001:** The cell mortality rate in the different *H. pluvialis* cell type cultures under different light intensities.

Strain	Major Cell Types	Light Intensity (μmol·m^−2^·s^−1^)	Cell Mortality Rate (ca. %)	References
CCMA-451	Motile cells	30	8	This work
CCMA-451	Motile cells	80	22	This work
CCMA-451	Motile cells	150	28	This work
CCMA-451	Nonmotile cells	30	4	This work
CCMA-451	Nonmotile cells	80	7	This work
CCMA-451	Nonmotile cells	150	8	This work

**Table 2 marinedrugs-17-00039-t002:** The average cell diameter, biomass content, and astaxanthin content of single *H. pluvialis* cysts in two different cultures at different light intensities.

Light Intensity (μmol·m^−2^·s^−1^)	Cysts in Motile Cell Cultures			Cysts in Nonmotile Cell Cultures		
	Average Cell Diameter (μm)	Biomass Content (ng·cell^−1^)	Astaxanthin Content (pg·cell^−1^)	Average Cell Diameter (μm)	Biomass Content (ng·cell^−1^)	Astaxanthin Content (pg·cell^−1^)
30	19.76 ± 2.66	0.93 ± 0.11	20.72 ± 3.49	28.71 ± 5.22	1.81 ± 0.07	38.49 ± 0.66
80	21.38 ± 3.44	1.13 ± 0.11	28.51 ± 3.26	33.07 ± 4.25	2.58 ± 0.15	86.21 ± 1.51
150	21.39 ± 3.85	1.31 ± 0.04	30.87 ± 0.56	33.97 ± 5.35	2.82 ± 0.14	108.61 ± 0.73

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
