# Peer review of "Differences between Motile and Nonmotile Cells of Haematococcus pluvialis in the Production of Astaxanthin at Different Light Intensities"

_marinedrugs, 2019, doi:10.3390/md17010039_

Round 1
Reviewer 1 Report
This study describes the astaxanthin production by two different types of cells (termed motile-and nonmotile cells) of marine H. pluvialis at the different light intensity. The work compares the differences in astaxanthin accumulation and cell mortality rate under photo-oxidative stress between the two cells. The aim seems to be the determination of the optimal conditions for astaxanthin preparation in cell types of H. pluvialis for the application of this antioxidant and anti-inflammatory biomolecule. According to that description, the manuscript is interesting and suitable for publication once some points will be modified or clarified.
In general, results are written as a mere description of the Figures and Tables, and some Figures and Tables contain much related data. Thus, this text might be shortened to avoid redundancy.
Other points to be adressed:
1) One important concern is related to the use of cysts throughout the manuscript. At line 48: cyst (aplanospores) is defined as a type of cells. Do the author mean that cysts are equivalent to nonmotile cells?, at Table 1, first line NIES-144, it seems that cysts are equivalent to nonmotile cells. According Table 2 they are not. In other words, cyst at Table 1 are a cell type, but according to Figure 2, they are contained in motile and nonmotile cells. Something is confusing. This should be clarified.
2) The intensity of the light in other studies is much high than 150, up to 950 mmoles photons/s. m2 at ref. 24 so that the cell mortality rates are not comparable. The increase from 80 to 150 moles photons/s. m2 increases very slightly the mortality from 7 to 8%. Thus the optimal conditions for astaxanthin accumulation is probably higher than 150 but lower than 950. This should be taken into account instead of the simple sentence (lines 228-229) “Unfortunately, exposure to excess high light intensity can cause high rates of cell mortality [16,24-228 26], thereby resulting in the low overall astaxanthin productivity”.
3) Lines 141-143: These lines state that the large differences in the astaxanthin growth rate between the two cell types exposed to photo-oxidative treatment are directly related to the accumulation of astaxanthin in nonmotile cells in short time to protect themselves from death under photo-oxidative stress.
In my opinion, the protective effect of astaxanthin is possibly contributing to the lower dead rate of nonmotile cells, but I do not think that this is the only reason. Likely, the previous transformation of motile to nonmotile cells during the 3 days stress conditions should create in the cells a different set of proteins and metabolites that can favor their growth rate after photoexposure in relation to “unprepared” motile cells. According the text, astaxanthin is accumulated inside the cyst? If so, how this molecule protects cellular biomolecules outside cysts from light generated ROS?. In summary, astaxanthin is probably not the only responsible of that pattern, and the paragraph in those lines would be rewritten to take into account this possibility.
Minor points:
Line 71: H. pluvialis
Line 77: in addition to mmoles photons/s. m2, could authors express the light intensity in any other unit such as PPFD, or watts/cm2 to help future readers not familiar with Einstein units?. Is there good correlation between the light source (a led) and sunlight related to the different wavenumbers distribution?
Table 1: The information of other strains included at Table 1 are not discussed at all. Some brief discussion would improve the manuscript.
Line 251: “astaxanthin ester [10,45]. Zhekisheva et al. [46] suggested that the high content of astaxanthin requires a large amount of lipids accumulation to maintain….”.
This is trivial, but which lipids do authors mean? Astaxanthin is an isoprenoid lipid itself, so that astaxanthin synthesis requires lipid precursors
Line 288: Are those cells considered nonmotile after just 3 days in the described conditions? Is nitrate also diluted 4 times? Clarify please.
Line 313: Astaxanthin is quantitated by absorbance at 492 nm (?1??1%=2220). Please, give reference of that value
Author Response
Response to Reviewer 1 Comments
Dear Reviewer 1:
Thanks for your comments concerning our manuscript ID marinedrugs-408446 entitled “Differences between motile and nonmotile cells of Haematococcus pluvialis in the production of astaxanthin at the different light intensity”. Those comments are all valuable and very helpful for revising and improving our paper. We have studied your comments carefully and have made revision. We have tried our best to revise our manuscript according to the comments. The main corrections in the paper and the responds to the comments are as following:
Point 1: One important concern is related to the use of cysts throughout the manuscript. At line 48: cyst (aplanospores) is defined as a type of cells. Do the author mean that cysts are equivalent to nonmotile cells?, at Table 1, first line NIES-144, it seems that cysts are equivalent to nonmotile cells. According Table 2 they are not. In other words, cyst at Table 1 are a cell type, but according to Figure 2, they are contained in motile and nonmotile cells. Something is confusing. This should be clarified.
Response: The word “cysts” in line 48 has been corrected as “haematocysts”, which are large red cells with a heavy resistant cell wall (Elliot 1934). The first line of Table 1 in the manuscript has been deleted.
Point 2: The intensity of the light in other studies is much high than 150, up to 950 mmoles photons/s. m2 at ref. 24 so that the cell mortality rates are not comparable. The increase from 80 to 150 mmoles photons/s. m2 increases very slightly the mortality from 7 to 8%. Thus the optimal conditions for astaxanthin accumulation is probably higher than 150 but lower than 950. This should be taken into account instead of the simple sentence (lines 228-229) “Unfortunately, exposure to excess high light intensity can cause high rates of cell mortality [16,24-228 26], thereby resulting in the low overall astaxanthin productivity”.
Response: The sentence “Unfortunately, exposure to excess high light intensity can cause high rates of cell mortality [16,24-26], thereby resulting in the low overall astaxanthin productivity” in lines 228-229 has been deleted and the sentence “The death of cells under photooxidative stress was attributed to the production of excess reactive oxygen species (ROS) [40,41], which can potentially react with major macromolecules (e.g. DNA, lipids, and protein) resulting in cellular damage [42].” in lines 230-232 has been revised as following “However, excess reactive oxygen species (ROS) which can potentially react with major macromolecules (e.g. DNA, lipids, and protein) produced in cells when prolong exposed to high light condition, resulting in cellular damage [40-42].”
Point 3: Lines 141-143: These lines state that the large differences in the astaxanthin growth rate between the two cell types exposed to photo-oxidative treatment are directly related to the accumulation of astaxanthin in nonmotile cells in short time to protect themselves from death under photo-oxidative stress.
In my opinion, the protective effect of astaxanthin is possibly contributing to the lower dead rate of nonmotile cells, but I do not think that this is the only reason. Likely, the previous transformation of motile to nonmotile cells during the 3 days stress conditions should create in the cells a different set of proteins and metabolites that can favor their growth rate after photoexposure in relation to “unprepared” motile cells. According the text, astaxanthin is accumulated inside the cyst? If so, how this molecule protects cellular biomolecules outside cysts from light generated ROS?. In summary, astaxanthin is probably not the only responsible of that pattern, and the paragraph in those lines would be rewritten to take into account this possibility.
Response: The sentence “to protect themselves under photooxidative stress” at the line 145 has been deleted and the sentences in line 141-143 have been revised as following “The large differences in the astaxanthin growth rate between the two cells in the early stage of induction indicated that once exposed to photooxidative stress, the nonmotile cells can rapidly synthesize and accumulate astaxanthin in a short time.”
Point 4: Line 71: H. pluvialis
Response: The wrong word “H. plvialis” has been corrected as “H. pluvialis”.
Point 5: Line 77: in addition to mmoles photons/s. m2, could authors express the light intensity in any other unit such as PPFD, or watts/cm2 to help future readers not familiar with Einstein units?. Is there good correlation between the light source (a led) and sunlight related to the different wave numbers distribution?
Table 1: The information of other strains included at Table 1 are not discussed at all. Some brief discussion would improve the manuscript.
Response: The light intensity unit in this manuscript is PPFD (μmol m-2 s-1). Illumination was provided from the side by 100% red (620-630 nm), 100% green (520-530 nm), and 100% blue (465-475 nm) LED plant grow lights (Xiamen Top-Succeed Photobiology Technology Co., Ltd, Xiamen, China). The details of light conditions have supplemented in the line 301.
The sentence “Hu et al. [25] reported that the highest cell mortality rate of the NIES-144 strains of H. pluvialis reached ca. 40% at the light intensity 250 μmol m-2 s-1, while Hata et al. [24] and Li et al. [26] used the same strains to obtain ca. 70% and 82% cell mortality rate under the light intensity of 950 μmol m-2 s-1 and 600 μmol m-2 s-1, respectively.” has been added in the manuscript (lines 236-240).
Point 6: Line 251: “astaxanthin ester [10,45]. Zhekisheva et al. [46] suggested that the high content of astaxanthin requires a large amount of lipids accumulation to maintain….”.
This is trivial, but which lipids do authors mean? Astaxanthin is an isoprenoid lipid itself, so that astaxanthin synthesis requires lipid precursors
Response: The word “lipids” in line 252 has been corrected as “oleic acid”.
Point 7: Line 288: Are those cells considered nonmotile after just 3 days in the described conditions? Is nitrate also diluted 4 times? Clarify please.
Response: The details of the preparation of nonmotile cells have supplemented as following: “A part of the five-day-old motile cell culture was diluted with phosphate-depleted medium containing 1.25 g L-1 NaCl at a ratio of 1:4, followed by cultivating under 30 μmol photons m-2 s-1 for 3 days to prepare nonmotile cells. To increase the quantity of nonmotile cells, cells that settled at the bottom of the glass columns were collected and washed with fresh aseptically medium two times to remove remaining motile cells.”
Point 8: Line 313: Astaxanthin is quantitated by absorbance at 492 nm (?1??1%=2220). Please, give reference of that value
Response: The reference has been added.
Reviewer 2 Report
The authors investigated improvement of astaxanthin production in Haematococcus pluvialis focused on cell-type and light intensity. They concluded that using nonmotile cells of H. pluvialis enhances the astaxanthin production stably.
This study is interesting and should provide a contribution to this field.
Overall, I think this paper could potentially be suitable for publication in Marine Drugs, but there are a few improvements that should be made.
1) You described that the collected motile and nonmotile cells were grown in stress media. What is “stress media”?
2) For Table 1, please describe why you need to show quoting other data in Table 1. I think that quotation is not necessary if only the comparison of the data obtained in this research in MS.
3) For Table 1, the values of cell mortality rate between motile cells of strain CCAP 34/12 and CCMA-451 grown under 150 mmol/m2/s light intensity are much different. How come?
4) Cyst formation of H. pluvialis would be important for enhancement of the astaxanthin production. The cysts do not divide and proliferate, don’t they? Is astaxanthin produced only in cysts, not in microzooid, macrozooid, nonmotile palmella cells?
5) For Figure 2, I feel uncomfortable with “astaxanthin growth rate.” Does astaxanthin grow??
6) I am not familiar with “productivity.” For example, in nonmotile cells, the productivities decreased over time in most cases, is it generally the case? High productivity is not necessarily better? Considering these productivities shown in Figure 2, is the second day the best?
7) Minor revisions are listed below.
1. Line 369: “Haematococcus pluvialis” should be italic.
2. Line 397: “mutant” should be plain.
3. You used the word “witness” in the MS. It may be better to replace it with other word.
Author Response
Dear Reviewer 2:
Thanks for your comments concerning our manuscript ID marinedrugs-408446 entitled “Differences between motile and nonmotile cells of Haematococcus pluvialis in the production of astaxanthin at the different light intensity”. Those comments are all valuable and very helpful for revising and improving our paper. We have studied your comments carefully and have made revision. We have tried our best to revise our manuscript according to the comments. The main corrections in the paper and the responds to the comments are as following:
Point 1: You described that the collected motile and nonmotile cells were grown in stress media. What is “stress media”?
Response: The words “stress media” has been revised as “N and P depletion medium” in the manuscript.
Point 2: For Table 1, please describe why you need to show quoting other data in Table 1. I think that quotation is not necessary if only the comparison of the data obtained in this research in MS.
Response: The quotation in Table 1 of the manuscript have been deleted.
Point 3: For Table 1, the values of cell mortality rate between motile cells of strain CCAP 34/12 and CCMA-451 grown under 150 mmol/m2/s light intensity are much different. How come?
Response: The difference in the cell mortality rate between motile cells of strain CCAP 34/12 and CCMA-451 may come from (1) the strains used were different and (2) the age of cells was different. As shown in Table 1, the cell mortality rate was much different between WT and MT2877 of the same strain NIES 144 under the same light intensity. For example, under the light intensity of 300 μmol photons m-2 s-1, the cell mortality rate in MT2877 and WT was 26% and 74%, respectively. In general, younger cells were more susceptible than the older ones. However, the author did not point out the age of motile cells (CCAP 34/12) in her study (Han et al., 2012).
Point 4: Cyst formation of H. pluvialis would be important for enhancement of the astaxanthin production. The cysts do not divide and proliferate, don’t they? Is astaxanthin produced only in cysts, not in microzooid, macrozooid, nonmotile palmella cells?
Response: The word “cysts” in line 48 has been corrected as “haematocysts”, which are large red cells with a heavy resistant cell wall, according to the reference (Elliot 1934). The life cycle of H. pluvialis goes through 4 stages: (1) vegetative stage under favorable environmental conditions; (2) encystment stage (vegetative to immature cyst cells); (3) maturation stage (immature to mature cyst cells); (4) germination stage: under favorable conditions mature cyst transform into vegetative cells (Kobayashi et al., 1997). Both the motile flagellate and nonmotile palmella cells are defined as vegetative cells. When environmental conditions become unfavorable for growth, the vegetative cells undergo transformation to form thick-walled cysts, accompanied by accumulation of astaxanthin. In most of cases, accumulation was accompanied with the inhibition of cell division. A small number of motile cells can also synthesis astaxanthin under nitrogen deficiency and low light conditions.
Point 5: For Figure 2, I feel uncomfortable with “astaxanthin growth rate.” Does astaxanthin grow??
Response: All “astaxanthin growth rate” (including figures) in the manuscript marinedrugs-408446 have been corrected with “astaxanthin content increase rate”.
Point 6: I am not familiar with “productivity.” For example, in nonmotile cells, the productivities decreased over time in most cases, is it generally the case? High productivity is not necessarily better? Considering these productivities shown in Figure 2, is the second day the best?
Response: The biomass productivity represents the biomass produced per day. The astaxanthin productivity represents the astaxanthin produced per day. They are both closely related to environmental factors like light, temperature and available nutrients. When expose to a favorable environment, the cells can maintain biomass productivity in a high level but cannot accumulate astaxanthin. When the environment become unfavorable for growth, the cells accumulate the astaxanthin. Since the cells are under stress condition in our study, the biomass productivity decreases over time. The faster the biomass productivity decline means the slower the increase in total biomass. The greater the astaxanthin productivity value means the more the cell produced astaxanthin. The two parameters reflect the differences in the production of biomass and astaxanthin between the two cells under stress conditions.
Point 6: Minor revisions are listed below.
1. Line 369: “Haematococcus pluvialis” should be italic.
2. Line 397: “mutant” should be plain.
3. You used the word “witness” in the MS. It may be better to replace it with other word.
Response: The words “Haematococcus pluvialis” in line 369 have been corrected in italic.
The words “mutant” in line 397 have been corrected in plain.
All words “witness” in the manuscript have been replaced with the word “observe”.